# Bidirectional Mediation Effects between Intratumoral Microbiome and Host DNA Methylation Changes Contribute to Stomach Adenocarcinoma

Kaile Yue,[a,b] Dashuang Sheng,[a,b] Xinxin Xue,[a,b] Lanlan Zhao,[a,b] Guoping Zhao,[a,b,c,d] Chuandi Jin,[a,b] Lei Zhang[a,b,c]

[a]Microbiome-X, National Institute of Health Data Science of China, Shandong University, Jinan, China
[b]Department of Biostatistics, School of Public Health, Cheeloo College of Medicine, Shandong University, Jinan, China
[c]State Key Laboratory of Microbial Technology, Shandong University, Qingdao, China
[d]CAS Key Laboratory of Computational Biology, Bio-Med Big Data Center, Shanghai Institute of Nutrition and Health, University of Chinese Academy of Sciences, Chinese Academy of Sciences, Shanghai, China

Kaile Yue, Dashuang Sheng, and Xinxin Xue contributed equally to this work. Author order was determined on the basis of contribution.

**ABSTRACT** The induction of aberrant DNA methylation is the major pathway by which *Helicobacter pylori* infection induces stomach adenocarcinoma (STAD). The involvement of the non-*H. pylori* gastric microbiota in this mechanism remains to be examined. RNA sequencing data, clinical information, and DNA methylation data were obtained from The Cancer Genome Atlas (TCGA) STAD project. The Kraken 2 pipeline was employed to explore the microbiome profiles. The microbiome was associated with occurrence, distal metastasis, and prognosis, and differential methylation changes related to distal metastasis and prognosis were analyzed. Bi-directional mediation effects of the intratumoral microbiome and host DNA methylation changes on the metastasis and prognosis of STAD were identified by mediation analysis. The expression of the *ZNF215* gene was verified by real-time quantitative PCR (RT-qPCR). A cell counting kit 8 (CCK8) cell proliferation experiment and a cell clone formation experiment were used to evaluate the proliferation and invasion abilities of gastric cells. Our analysis revealed that *H. pylori* and other cancer-related microorganisms were related to the occurrence, progression, or prognosis of STAD. The related methylated genes were particularly enriched in related cancer pathways. *Kytococcus sedentarius* and *Actinomyces oris*, which interacted strongly with methylation changes in immune genes, were associated with prognosis. Cell experiments verified that *Staphylococcus saccharolyticus* could promote the proliferation and cloning of gastric cells by regulating the gene expression level of the *ZNF215* gene. Our study suggested that the bi-directional mediation effect between intratumoral microorganisms and host epigenetics was key to the distal metastasis of cancer cells and survival deterioration in the tumor microenvironment of stomach tissues of patients with STAD.

**IMPORTANCE** The burgeoning field of oncobiome research declared that members of the intratumoral microbiome besides *Helicobacter pylori* existed in tumor tissues and participated in the occurrence and development of gastric cancer, and the methylation of host DNA may be a potential target of microbes and their metabolites. Current research focuses mostly on species composition, but the functional genes of the members of the microbiota are also key to their interaction with the host. Therefore, we focused on characterizing the species composition and functional gene composition of microbes in gastric cancer, and we suggest that microbes may further participate in the occurrence and development of cancer by influencing abnormal epigenetic changes in the host. Some key bioinformatics analysis results were verified by *in vitro* experiments. Thus, we consider that the tumor microbiota-host epigenetic axis of gastric cancer microorganisms and the host explains the mechanism of the microbiota participating in cancer occurrence and development, and we make some verifiable experimental predictions.

**Editor** Zhenjiang Zech Xu, 南昌大学

Address correspondence to Lei Zhang, zhanglei7@sdu.edu.cn, or Chuandi Jin, jinchuandi@sdu.edu.cn.

The authors declare no conflict of interest.

**KEYWORDS** stomach cancer, gastrointestinal microbiome, tumor microenvironment, host-microbe interactions, DNA methylation

Gastric cancer (GC) is the fifth most common cancer and the fourth most common cause of cancer deaths globally (1), with adenocarcinoma being a key type. Microbes, with *Helicobacter pylori* as the core and predominant key risk factor, participate in the occurrence and development of gastric cancer (2). The burgeoning field of oncobiome research (3), focused on the interplay between the human microbiome and cancer development, has proven that tumor-type-specific intracellular bacteria are mostly present in both cancer and immune cells as an indispensable part of the tumor microenvironment (4, 5). The cancer-specific microorganisms in tissues and blood can be clinically informative for cancers, especially gastrointestinal cancers, and promote the distal metastasis of cancer cells (6), which highlights the importance and variability of the intratumoral microbiome in cancer occurrence and progression.

Increasing evidence suggests that microbes and microbial function genes could promote carcinogenesis but are insufficient to cause cancer (7), which requires mediators to locally or remotely regulate tumor progression and therapeutic effects. Abnormal methylation, as an early event in gastric cancer, is considered an "epigenetic field for cancerization," which is the potential target of microbes and their metabolites (8). In the last few years, many efforts have focused on investigating how gut microbiota metabolism regulates the concentrations and/or activities of metabolites in the host that lead to altered epigenetic marks, termed the microbiota-host epigenetic axis. Exposure to the commensal microbiota induces localized DNA methylation changes in "early sentinel" response genes (9). The induction of aberrant DNA methylation, with a greater effect than genetic changes in normal stomach tissue, is the major pathway by which *H. pylori* infection induces gastric cancer (10). The regulation of the methylation modification of key genes in the host by colorectal-cancer-related microbes explains the pathogenesis and potential intervention targets of colorectal cancer (11). There are also tumor-enriched taxa and methylation changes in gastric cancer that have been used as biomarkers and intervention targets and have an impact on cancer initiation and progression and the response to therapy, including cancer immunotherapy and drug resistance (12, 13). K. J. Thompson et al. examined both the presence of microbes and host expression from the same tissue/sample preparations by using The Cancer Genome Atlas (TCGA) breast cancer data and associated the microbiota with tumor expression profiles (14). It is worth exploring whether there is an interaction between the intratumoral microbiota and host methylation changes that could contribute to the development and prognosis of gastric cancer.

At present, most studies on intratumoral bacteria have focused on differences in the phylogenic composition, whereas the microbial genetic composition could further facilitate the exploration of the mechanism of the microbiome participating in gastric cancer. In this study, we extracted microbial reads from TCGA STAD (stomach adenocarcinoma) RNA sequencing data and explored the microbiome profiles and potential mechanism of microbes involved in the occurrence, development, and prognosis of gastric cancer. The DNA methylation profile associated with the distal metastasis of STAD was also analyzed, and whether the microbial features and methylation features bidirectionally mediate each other's contribution to the development and prognosis of gastric cancer was explored. Also, we further verified the key findings by *in vitro* experiments.

## RESULTS

**Members of the intratumoral microbiota are related to the occurrence, distal metastasis, and prognosis of STAD.** A total of 10.8% of the RNA sequencing reads from The Cancer Genome Atlas (TCGA) STAD project were defined as nonhuman reads, 18.8% of the nonhuman reads after quality control were mapped using Kraken 2 after decontamination, and 1,236 genera and 4,597 species were identified (see Tables S1 and S2 in the supplemental material). After decontamination processes (Table S3), totals of 1,168 genera and 4,004 species (Tables S4 and S5) were retained for the

following analysis. These genera had 62.3% and 48.06% intersections with those identified using Kraken and SHOGUN, respectively (Fig. S1A) (6).

The microbiome profile of STAD tissues was significantly different from that of adjacent normal tissues. Reduced alpha diversity was observed in the STAD tissues compared with that in the adjacent normal tissue ($P < 0.05$) (Fig. 1A). Beta diversity values were significantly different between groups ($P = 0.008$) (Fig. 1B). Totals of 120 differentially abundant taxa (Fig. S1B and C) and 55 differentially expressed functional genes (false discovery rate [FDR] of <0.10) (Table S9) were identified by the linear discriminant analysis effect size (LEfSe) algorithm. The expression level of the *fliZ* gene (KEGG entry: K02425) was increased significantly in STAD tissues (Fig. 1C). PICRUSt results showed that 48 Kyoto Encyclopedia of Genes and Genomes (KEGG) pathways were differentially enriched between cancerous tissues and adjacent normal tissues (Fig. 1D). The functional categories enriched in tumor tissues were related to the pathogenicity of bacteria and amino acid metabolism and synthesis. According to Boruta-RandomForest analyses, 16 relevant microbe features (Fig. 1F) among the differentially abundant members of the microbiota confirmed the importance of the intratumoral microbiota for the diagnostic ability for STAD and achieved a remarkable performance (area under the receiver operating characteristic [ROC] curve [AUC] = 0.931; area under the precision-recall [AUPR] curve = 0.930) (Fig. 1E), such as *Helicobacter pylori*, *Staphylococcus cohnii*, *Brachybacterium faecium*, human mastadenovirus C (HAdV-C), and *Micrococcus luteus*.

Our next focus was to determine whether the intratumoral microbiota played roles in the distal metastasis of STAD. We found that the majority (22/23) of differential taxa between the distant metastasis stage of cancer has spread to other parts of the body (M1) and cancer has not spread to other parts of the body (M0) or metastasis cannot be measured (MX) groups had higher abundance in the M1 group (Fig. 2A). Moreover, the node degree of the whole microbial cooccurrence network for the M1 group was higher than that for the M0/MX group, which suggested that the microbial interactions in STAD patients with distal metastasis were more complex (Fig. 2B and C; Tables S6 and S7). Five hub genera, *Acinetobacter*, *Halomonas*, *Vibrio*, *Mesorhizobium*, and *Staphylococcus*, were identified in the M1 network using GMrepo. There were 13 functional genes differentially expressed between the M0/MX and M1 groups (FDR of <0.10) (Table S9), among which the *fliZ* gene and the *yafN* gene were differentially expressed in the M0/MX group versus the M1 group (Fig. 2D and E), and high expression levels of the latter were related to poor prognoses (Fig. 3E).

Finally, the potential prognostic value of the intratumoral microbiota in STAD was assessed. Univariant Cox analysis identified 278 species as potential prognostic factors for overall survival (OS) (Table S8). Among these, *Kytococcus sedentarius*, *Brachybacterium avium*, *Dolosigranulum pigrum*, and *Staphylococcus cohnii* (Fig. 3A) showed not only differential abundances between the M1 and M0/MX groups but also potential prognostic value for STAD. Furthermore, a prognostic model consisting of 13 potentially prognostic species (3 protective factors and 10 risk factors) (Fig. 3B) selected by LASSO (least absolute shrinkage and selection operator)-Cox analysis performed well in predicting the rate of survival of patients, and the risk score calculated by it showed significant differences among different survival outcomes (Fig. 3C and D). A total of 11 functional genes were found to be associated with the prognosis of patients with STAD (FDR of <0.10) (Table S9). High expression levels of the *yafN* gene and enrichment for the "sphingolipid metabolism" pathway were associated with poor prognoses (Fig. 3E and F). Taken together, members of the intratumoral microbiota were found to be involved in the occurrence, distal metastasis, and prognosis of STAD and showed likely diagnostic and prognostic value as well.

**Aberrant DNA methylation is involved in the distal metastasis and prognosis of STAD.** A total of 4,799 differentially methylated probes (DMPs) were identified between the M0/MX and M1 groups (|log$_2$ fold change| of >0.05). For the majority of the DMPs (2,774 of 4,799), the methylation levels were higher in the M1 group than in the M0/MX group (Fig. 4A), and their corresponding genes were particularly enriched in 3 pathways (Fig. 4B). A total of 14 candidate differentially methylated regions (DMRs), encompassing 197 probes, were identified and hypermethylated in the M1

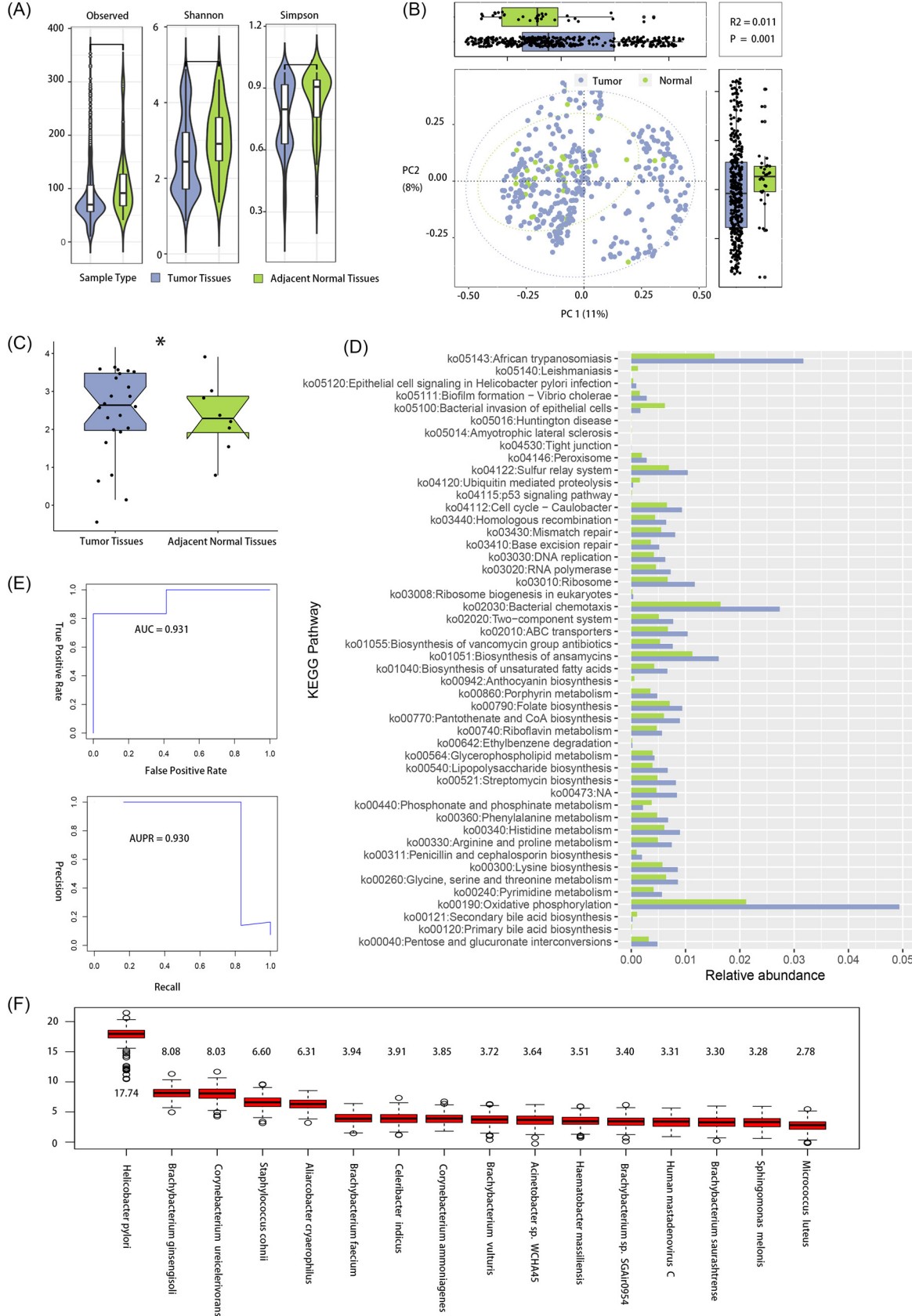

**FIG 1** The microbiomes of tumor tissue and adjacent normal tissue are significantly different. (A and B) Bacterial alpha diversity (A) measured by the observed, Shannon, and Simpson indices and beta diversity (B) measured by Bray-Curtis distances in tumor tissues

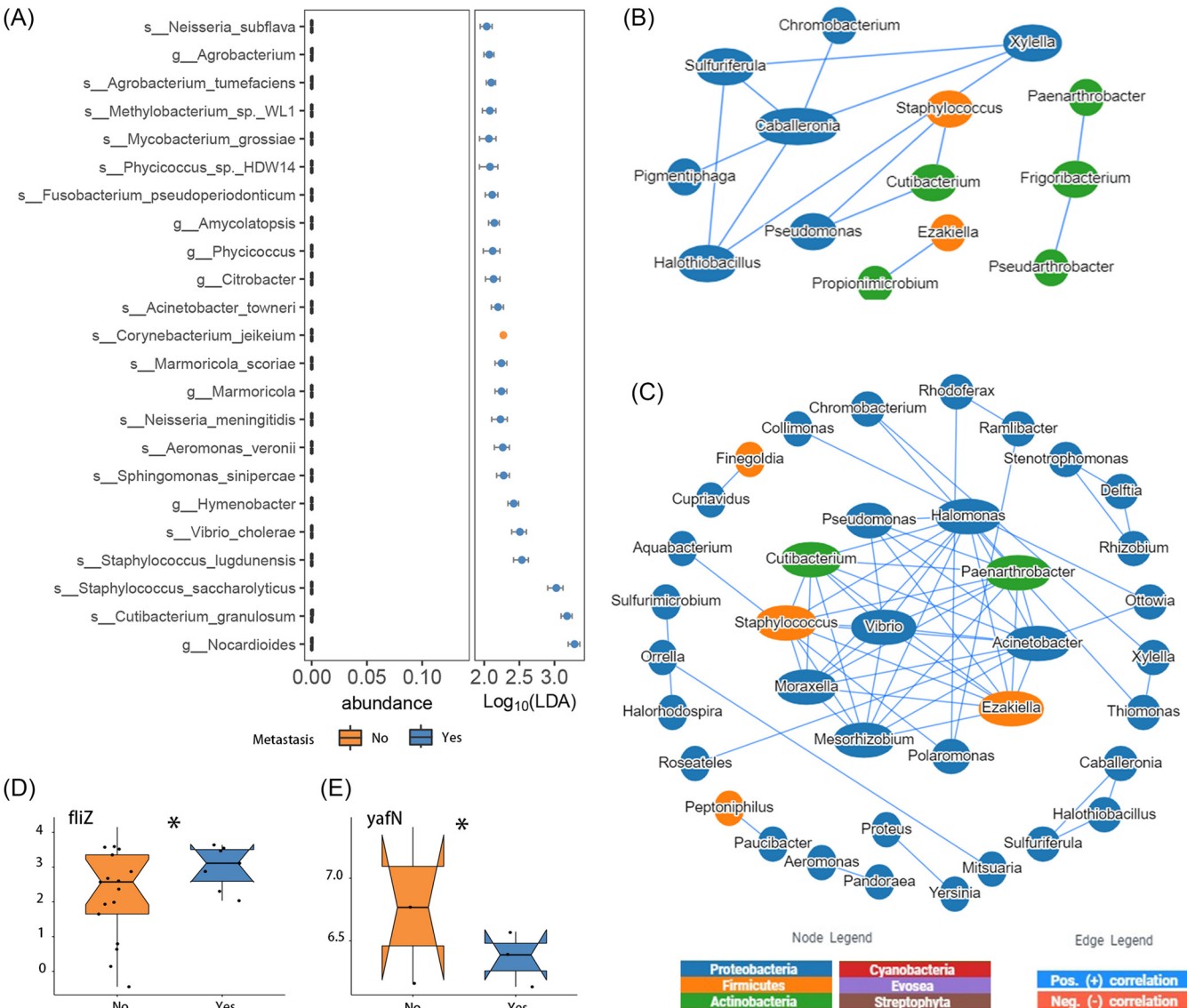

**FIG 2** Changes in intratumoral microbiome profiles are related to the development of gastric cancer. (A) Differences in the abundances of taxa between the M0/MX and M1 groups. LDA, linear discriminant analysis. (B and C) Categorical network analysis of the intratumoral bacteria in the M0/MX (B) and M1 (C) groups based on correlation analysis. Nodes are colored according to their phylum affiliations and sized according to their degrees. The blue lines between two nodes indicate positive correlations, whereas the red lines indicate negative correlations. (D and E) Evaluation of differences in the *fliZ* (D) and *yafN* (E) genes between the M0/MX and M1 groups.

group (Fig. 4C; Fig. S2). The methylation levels of the altered genes according to the DMRs were negatively correlated with their expression levels (except for the *MIAT* gene). Based on these methylated genes, seven interaction hot spot modules were identified from the interaction network (Fig. S3). The gene lists of these modules were enriched in "pathways in cancer," "Th1 and Th2 cell differentiation," and the "*NOTCH* signaling pathway," etc. (Fig. 4D). The *CHGB*, *HIF3A*, *VWA5B2*, and *TBX6* genes were not only abnormally methylated genes but also differentially expressed genes (DEGs) between the M0/MX and M1 groups (Fig. 4E).

**FIG 1** Legend (Continued)

and adjacent normal tissues. (C) Evaluation of differences in K02425 (*fliZ* gene) between tumor and normal tissues. (D) Functional KEGG pathway enrichment analysis of microbiota function genes between tumor tissues and normal gastric tissues (FDR of <0.05). NA, not applicable. (E) AUC (top) and AUPR curve (bottom) to measure the model performance of 16 relevant microbe features in primary tumor tissues in the validation set. (F) Z-scores of the 16 relevant microbe features in primary tumor tissues.

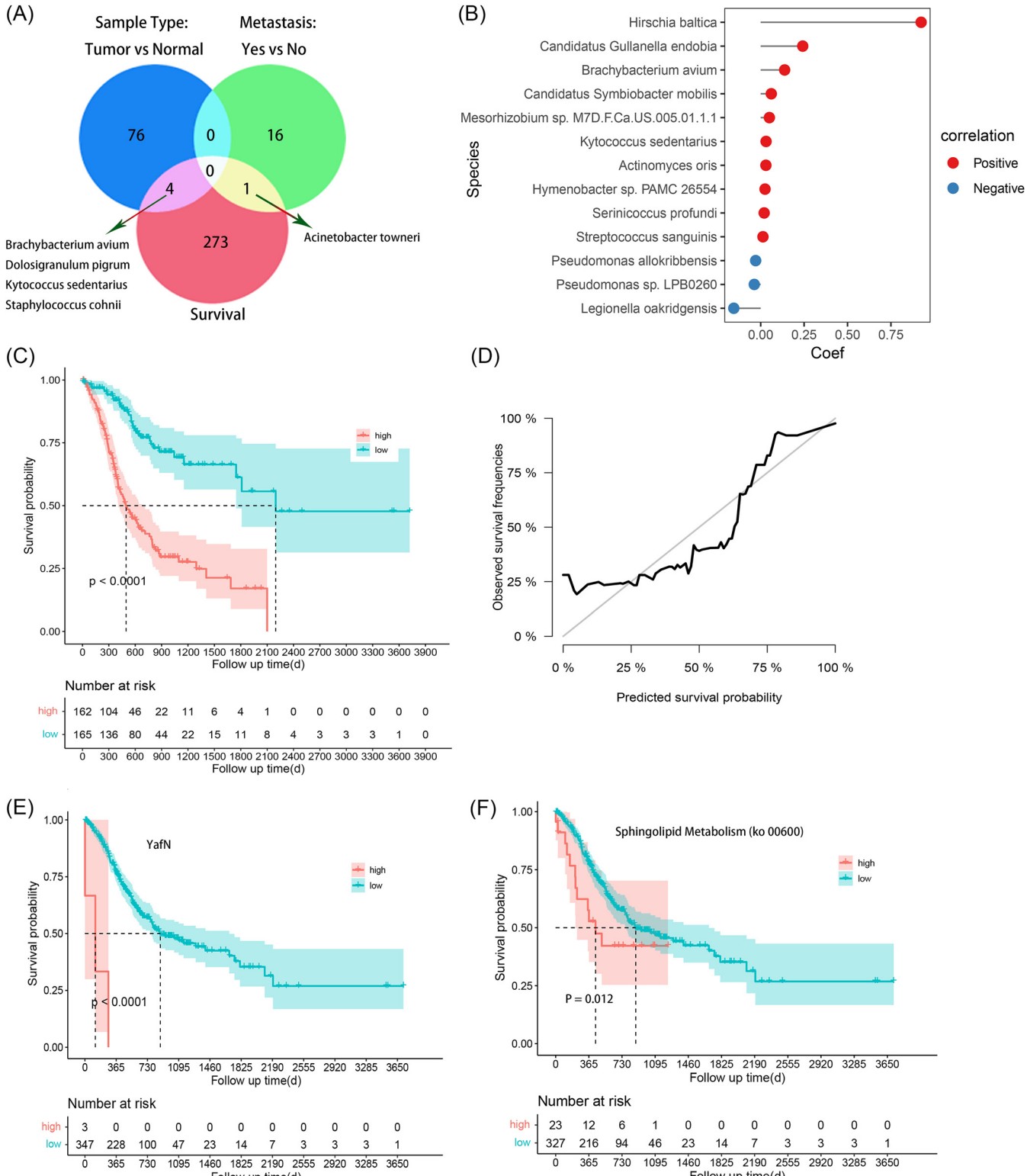

**FIG 3** Changes in the intratumoral microbiome have prognostic value for the survival of gastric cancer patients. (A) Venn plot of differentially abundant microbes for tumor versus normal tissues (top left), the M0/MX group versus the M1 group (top right), and prognosis (bottom) identified by the LEfSe algorithm or CoxPH. (B) Partial regression coefficients of 13 species with prognostic value identified by LASSO-Cox regression. (C) Kaplan-Meier estimates of overall survival according to the risk score calculated using 13 prognostic microbes. The *P* values were calculated using the log rank test. (D) Survival probability predicted using 13 prognostic microbes and true survival probability for STAD patients within 3 years in the validation set. (E) Overall survival according to the level of K19161 (*yafN* gene) estimated by Kaplan-Meier analysis. The *P* values were calculated using the log rank test. (F) Overall survival according to the level of ko00600 estimated by Kaplan-Meier analysis. The *P* values were calculated using the log rank test.

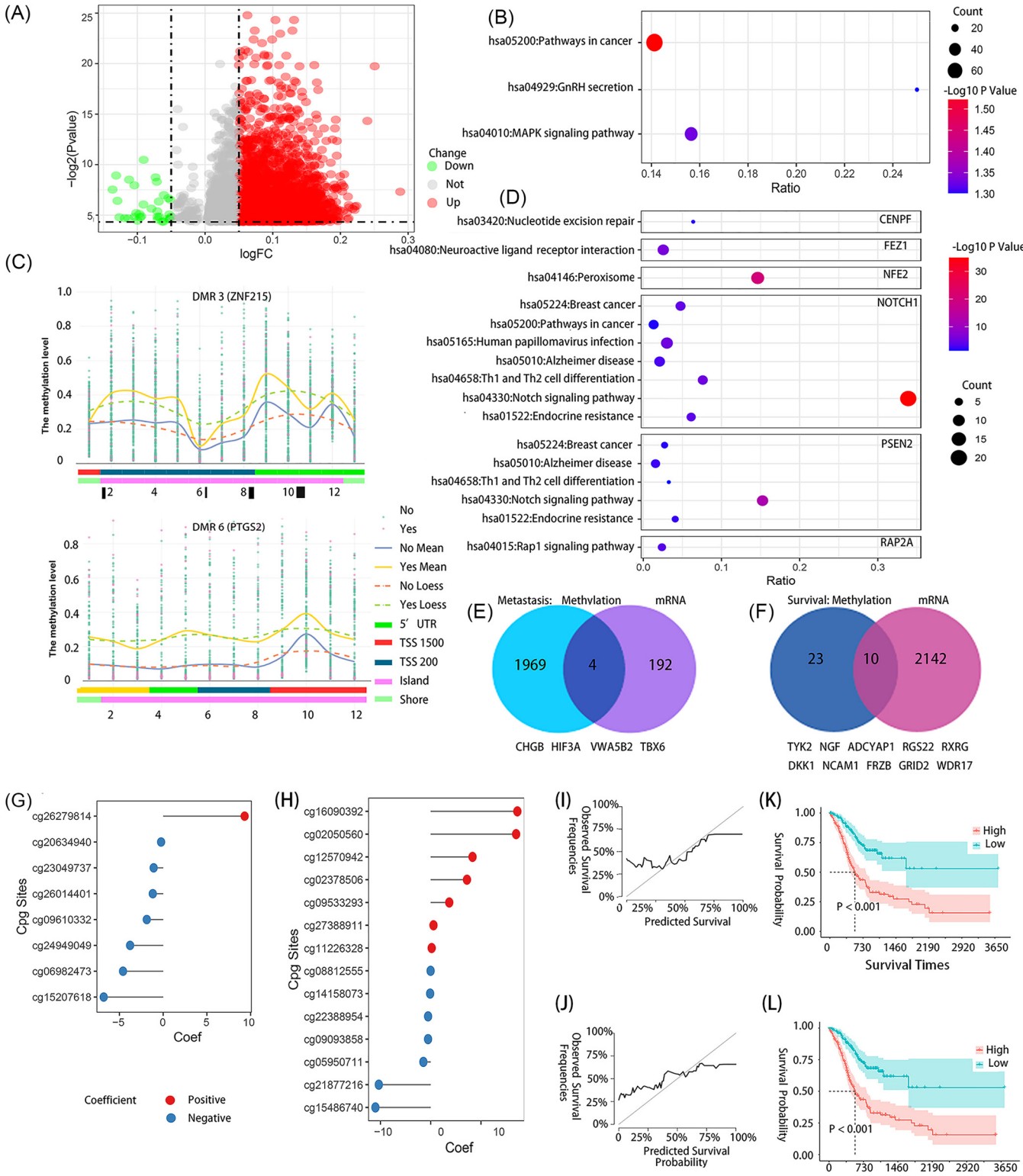

**FIG 4** Aberrant DNA methylation changes between the M0/MX and M1 groups or patient survival. (A) Volcano plot of DMPs between the M0/MX and M1 groups. DMPs with a log fold change (logFC) of >0.05 were considered high-methylation sites in the M1 group, and those with a log fold change of <0.05 were considered low-methylation sites. (B) Functional KEGG pathway enrichment analysis of the methylated gene of DMPs. GnRH means gonadotropin hormone-releasing hormone. (C) Information on the CpGs involved in DMR 3 and DMR 6. The bottom shows the CpG annotations, including relationships to transcription start sites (TSS), CpG islands, or other regions. Loess, locally estimated scatterplot smoothing; UTR, untranslated region. (D) KEGG pathway enrichment analysis for each interaction hotspot modules. (E) Genes with methylation changes (left) and expression level changes (right) related to metastasis. (F) Genes with methylation changes (left) and expression level changes (right) related to the prognosis of patients. (G) Partial regression coefficients of 8 DMPs

Totals of 12 DMPs and 45 DMRs were identified as potential prognostic methylation biomarkers (Table S10) based on univariate Cox analysis (FDR of <0.25). Multivariate Cox analysis further showed that the prognostic model comprising 8 DMPs or 14 DMRs ($P <$ 0.05) (Fig. 4G and H) selected by LASSO-Cox regression could well predict the rate of survival of patients (Fig. 4I and J; Table S11). The risk scores calculated by the prognostic signatures of DMPs or DMRs showed significant differences among different survival outcomes ($P <$ 0.001) (Fig. 4K and L). Besides, a total of 2,152 DEGs were identified as potential prognostic gene biomarkers. A Venn diagram showed that both the methylation and expression of 10 genes have potential prognostic value for patients with STAD (Fig. 4F). Moreover, 4 of these 10 genes were found to be associated with the distal metastasis of STAD as well. Generally speaking, distal metastasis in STAD patients may be related to more hypermethylation genes, and some aberrant methylation sites and regions have potential predictive abilities for prognosis.

**Bi-directional mediation between the microbiome and host methylation contributes to STAD.** Of the 39 features associated with the distal metastasis of STAD, there were 21 microbial features and 18 methylated features. The correlation heatmap showed highly significant correlations between the methylated features and the microbial features (Fig. S4A). A total of 20 potential mediation linkages related to the distal metastasis of STAD were revealed by bidirectional mediation analysis (Table S12). The interaction network showed that the methylated features tend to promote the distal metastasis of STAD by affecting the diversity of the intratumoral microbiota. There was a potential bidirectional mediation effect between the abundance changes in *Staphylococcus saccharolyticus* and the methylation changes in the *ZNF215* gene, and *S. saccharolyticus* also mediated the effect of methylation changes in the *PTGS2* gene on metastasis (Fig. 5A).

There were 70 features that could be regarded as potential prognosis markers for STAD patients, including 13 microbial features and 57 methylated features. The microbial features were generally associated with the methylated features (Fig. S4B and C). Survival mediation analyses provided 51 potential mediation associations between the microbial features and the methylated features (Table S13). The microbe-methylation potential casual interaction networks are shown in Fig. 5B. For instance, *K. sedentarius* may contribute to the prognosis of STAD patients by affecting nine methylation features (34.9%). Meanwhile, it could also mediate the effects of these nine methylation characteristics on prognosis (45.1%). *Actinomyces oris* could contribute to the prognosis of STAD patients by affecting 6 methylation features (23.3%), and it also mediated the effect of 2 of these 6 methylation features and the other 2 methylation features on prognosis (34.2%). There may be a bi-directional mediation effect between *Streptococcus sanguinis* and the hypermethylation of the cg12570942 site of the *DTYMK* gene. In summary, the complex interaction between methylation changes and microbial characteristics may promote distal metastasis and affect the prognosis of STAD patients.

***S. saccharolyticus* promotes cancer cell metastasis by regulating the expression level of the *ZNF215* gene.** To further explore the mechanism underlying the observed associations, we used three human gastric cell lines to verify the effect of the interaction between *S. saccharolyticus* and the *ZNF215* gene on the metastasis of cancer cells.

By comparing group 1 and group 6, we could verify the correlation among *S. saccharolyticus*, cancer cell metastasis, and the expression of *ZNF215*. Real-time quantitative PCR (RT-qPCR) results showed that the expression levels of the *ZNF215* gene in the GES-1 cell line and the HGC-27 cell line were increased in group 1 compared to group 6, but the opposite result was found for the AGS cell line (Fig. 5C). The results of cell clone formation experiments and cell counting kit 8 (CCK8) experiments showed that

**FIG 4** Legend (Continued)
with prognostic value identified by LASSO-Cox regression. (H) Partial regression coefficients of 12 DMR sites with prognostic value identified by LASSO-Cox regression. (I) Survival probability predicted using 8 prognostic DMPs (left) and true survival probability for STAD patients within 5 years (right) in the validation set. (J) Survival probability predicted using 12 prognostic DMRs (left) and true survival probability for STAD patients within 5 years (right) in the validation set. (K) Kaplan-Meier estimates of overall survival according to the risk score calculated using 8 prognostic DMPs. (L) Kaplan-Meier estimates of overall survival according to the risk score calculated using 12 prognostic DMRs.

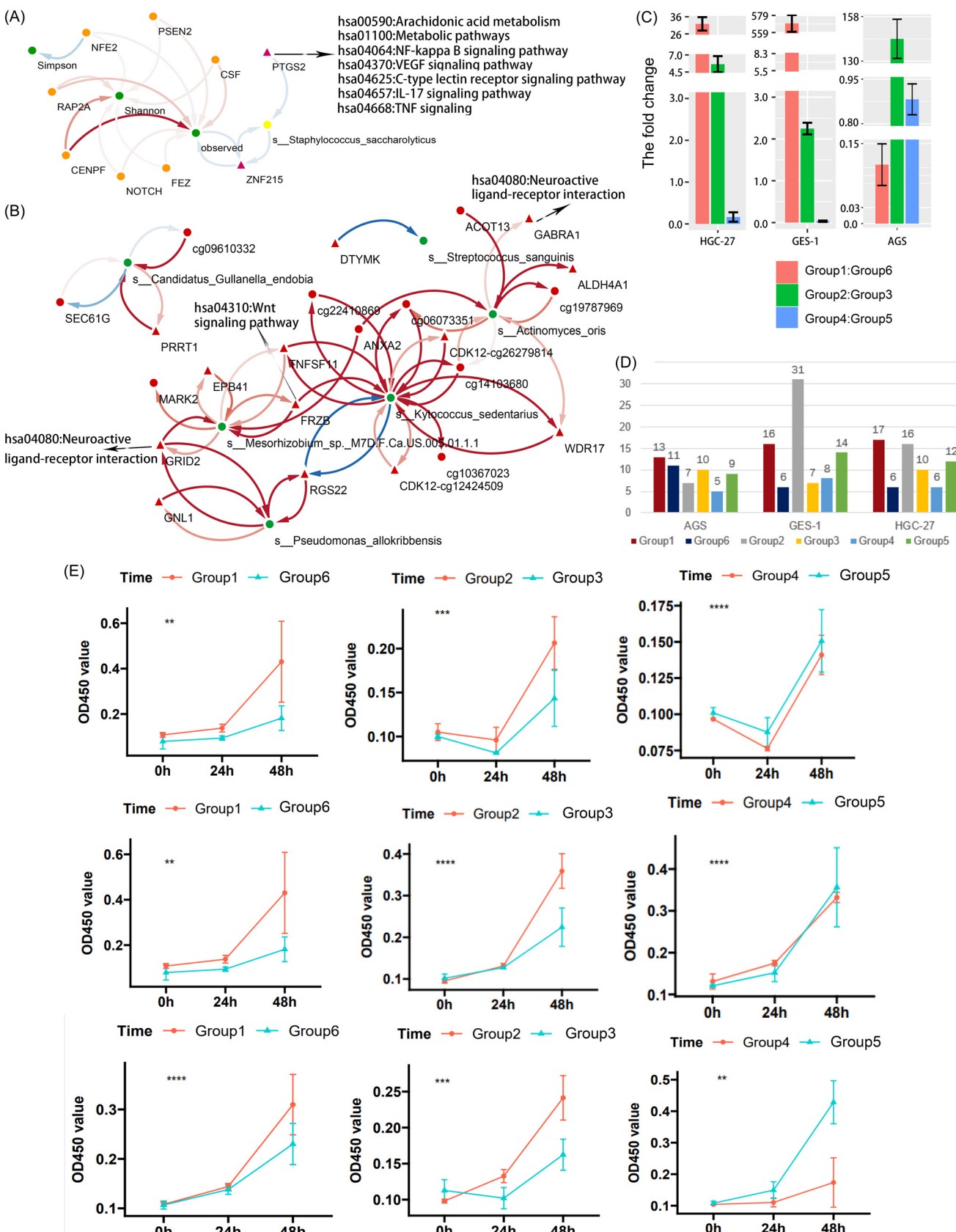

**FIG 5** Bi-directional mediation effects between host epigenetic changes and microbial features contributing to STAD and results of the cell experiments. (A) Network of interactions between potential mediation linkages related to distal metastasis in STAD patients. Nodes represent microbial

the number of cell clones in group 1 and the number of living cells in group 1 within 48 h were higher than those in group 6 (Fig. 5D and E). These results indicated that *S. saccharolyticus* could modify the gene expression level of the *ZNF215* gene and further promote the metastasis of gastric cancer cells and normal gastric cells.

By comparing group 4 and group 5, the correlation between the decreased expression level of the *ZNF215* gene and the distal metastasis of cancer cells could be verified. RT-qPCR results showed that the expression levels of the *ZNF215* gene in the three cell lines were increased in group 5 (Fig. 5C). The results of cell clone formation experiments and CCK8 experiments showed that the number of cell clones in group 5 and the number of living cells in group 5 within 48 h were higher than those in group 4 (Fig. 5D and E). These results indicated that the increased expression of *ZNF215* could promote the metastasis of gastric tissue cells.

By comparing group 2 and group 3, the results showed that *S. saccharolyticus* could further promote cancer cell metastasis by regulating the expression level of *ZNF215*. RT-qPCR results showed that the expression levels of the *ZNF215* gene in three cell lines were increased in group 2 (Fig. 5C). CCK8 experimental results showed that the number of living cells in group 2 was higher than that in group 3 within 48 h (Fig. 5E). The cell clone formation experimental results showed that the number of cell clones formed in group 2 was higher than that in group 3 for the GES-1 cell line and the HGC-27 cell line, but the number of cell clones formed in group 2 was lower than that in group 3 for the AGS cell line (Fig. 5D). The above-described results verified the indirect effect observed in the mediation analysis. Generally speaking, the above-described experimental results provide support for our data analysis results to a certain extent.

## DISCUSSION

Accumulating recent studies have focused on the roles of the intratumoral microbiota in the tumor initiation, progression, and prognosis of various cancers (10, 11, 14–16). In this study, intratumoral microbiome profiles were drawn for the microbiome taxon composition and functional gene composition. Members of the intratumoral microbiota were related to the occurrence, distal metastasis, and prognosis of STAD, which exhibited great diagnostic and prognostic value. Our results indicated that distal metastasis in STAD patients may be related to the presence of more hypermethylation genes, and some aberrant methylation sites and regions have predictive ability for prognosis. Furthermore, the complex mediation effects between methylation changes and microbial characteristics may promote distal metastasis and affect the prognosis of STAD patients, and cell experiments verified that *S. saccharolyticus* could further promote cancer cell metastasis by regulating the expression level of the *ZNF215* gene.

The microbial community identified by our analysis showed greater diversity than what was previously reported (17). We noted decreased diversity and richness in peritumoral and tumoral microhabitats than in adjacent normal tissues, which was consistent with the results of Liu et al. (18). Moreover, specific intratumoral microbial taxa, functional genes, and pathways involved in STAD were further identified. Our research showed that *H. pylori* was the most relevant microorganism in gastric cancer, and many studies have contributed to a deep understanding of various pathogenicity mechanisms of *H. pylori* involved in gastric cancer (19, 20). In addition, *B. faecium*, HAdV-C, and *M. luteus* were found to be present at higher abundances in STAD tissues than in adjacent normal tissues. *B. faecium* is capable

**FIG 5** Legend (Continued)

characteristics (green) and methylation changes, with orange circles representing methylation modules and red triangles representing DMRs. VEGF, vascular endothelial growth factor; IL-17, interleukin-17; TNF, tumor necrosis factor. (B) Network of interactions between potential mediation linkages related to the prognosis of STAD patients. Nodes represent microbial characteristics (green) and methylation changes (red), with triangles indicating that the methylation level of the CpG site or region was negatively correlated with the expression level of the gene. The color depth of the edge indicates the proportion of indirect effects to total effects; red indicates a positive effect, and blue indicates a negative effect. The source and track of the arrow are the independent variable and the mediator, respectively. (C) Fold changes in the expression levels of the *ZNF215* gene between groups of the three cell lines. Means and standard errors of the means (SEM) are shown. (D) Colony formation assays to detect cell proliferation in the three cell lines. (E) CCK8 assays to determine the cell viability of the AGS (first line), GES-1 (second line), and HGC-27 (third line) cell lines, including group 1 versus group 6 (first column), group 2 versus group 3 (second column), and group 4 versus group 5 (third column).

of degrading uric acid, which may affect the ability of uric acid to scavenge free radicals and fight cancer as a natural antioxidant in the human body, and fermenting cellobiose, glucose, maltose, and mannose (21, 22). HAdV-C always leads to respiratory tract infections in children and further enters and colonizes the gastrointestinal tract as an opportunistic pathogen (23). The effects of *M. luteus* on cancer have been inconsistent among different studies (24, 25). It is worth noting that the expression of microbial functional genes and pathways related to microbial pathogenicity, toxicity, and invasiveness were upregulated or enriched in STAD tissues, such as the *LDH* gene involved in glycolysis/gluconeogenesis and central carbon metabolism in the cancer pathway, the *ppaX* gene participating in the oxidative phosphorylation pathway, and the *sabA* gene involved in epithelial cell signaling in the "*Helicobacter pylori* infection" pathway, the "lipopolysaccharide biosynthesis" pathway, the "bacterial chemotaxis" pathway, and the "bacterial invasion of epithelial cells" pathway. Elevated levels of amino acids in the microenvironment of tumor tissue are factors that contribute to carcinogenesis (26). The expression level of the *fliZ* gene of bacteria is increased significantly in cancer tissues, especially in patients with distal metastasis of cancer cells. Previous studies have shown that the *fliZ* gene can regulate the pathogenicity of bacteria (27). KEGG analysis of microbial genes showed that cancer tissues were indeed enriched in amino acid metabolic pathways. The above-described results suggest that there were some opportunistic pathogens in the tumor microenvironment, which on the one hand increased the ability for self-expansion and on the other hand accelerated the metabolism of amino acids and other substances, thus promoting pathological changes in STAD tissue.

Metastasis accounts in part for the high rate of mortality from gastric cancer (28). A previous study showed that bacteria within breast tumor cells contribute to metastasis (5). Our research suggested that more complex microbial interactions in tumor tissue may promote the distal metastasis of STAD. The microbial cooccurrence network centralities suggested potentially important roles of *Acinetobacter*, *Staphylococcus*, and other common gastrointestinal cancer marker taxa in gastric cancer progression (29–31). A previous study showed that *Acinetobacter* was related to metabolites and may promote stomach inflammation in patients with gastric cancer (32). *Staphylococcus* is also the dominant bacterium of the stomach microbiota, and it is a urease-producing organism. Recently, it was reported that some species of *Staphylococcus* are prevalent in patients with dyspepsia (33).

The bacteria enriched in tumors could well predict the rate of survival of patients within 3 years, such as *K. sedentarius*, *S. sanguinis*, and *A. oris*. *K. sedentarius* is an opportunistic pathogen that is involved in the pathways of metabolism and genetic information processing (34). *Streptococcus* species have been shown in recent research to elicit potent Th1 immune responses and to promote CD8$^+$ T-cell responses that may decrease cancer development (35). Significant enrichments of *Streptococcus anginosus* and *Streptococcus constellatus* in GC tumor tissues were observed; however, *Streptococcus parasanguinis* and *Streptococcus sanguinis* were underrepresented in cases of gastric intestinal metaplasia compared to controls (36, 37). However, our study showed that *Streptococcus sanguinis* was a risk factor for prognosis, and this inconsistency suggested that although its abundance was low in cancer tissue, its dangerous effect on prognosis should not be ignored. During the formation of oral biofilms, *A. oris* is one of the most important and earliest-settled groups, which may descend to and colonize the stomach as a member of the tumor bacteria (38). *yafN*, a microbial functional gene encoding an antitoxin-toxin operon, was related not only to the survival of patients but also to whether cancer cells metastasize (39).

As increasing evidence has suggested that abnormal methylation in the host is a potential target of microbes and their metabolites, this prompted us to further explore whether host aberrant methylation could mediate the microbial impact on host phenotypes. The CpG site of the *FGFR2* gene was hypomethylated in the metastatic group, and the overexpression of the FGFR2 protein in gastric cancer indicates a poor prognosis (40). The hypomethylation level in specific regions of the *DKK1* gene is a protective factor for survival, and research has shown that DKK1 promotes tumor immune evasion

and impedes anti-PD-1 treatment by inducing immunosuppressive macrophages in GC (41). Our mediation analysis suggested that *S. saccharolyticus* may contribute to the distal metastasis of cancer cells by methylation changes in the *ZNF215* gene, and *S. saccharolyticus* may also mediate the effect of methylation changes in the *ZNF215* gene and the *PTGS2* gene on the distal metastasis of cancer cells. The methylation levels of the *ZNF215* gene and the *PTGS2* gene were negatively correlated with their expression levels. Furthermore, the *ZNF215* gene can be used as a diagnostic and prognostic biomarker for basal-like breast cancer and acute myeloid leukemia (42, 43). Among them, the view that *S. saccharolyticus* can further promote the proliferation and cloning of gastric cells by regulating the gene expression level of *ZNF215* has been verified by cell experiments in this paper, which increased the reliability of our data analysis results. Multiple follow-up studies revealed that *PTGS2* (*COX-2*) levels are elevated in many solid tumors, including stomach, esophagus, colorectum, liver, pancreas, lung, and breast cancers (44), and *PTGS2* might be employed as an adjunctive therapeutic target for the reversal of chemoresistance in a subset of cisplatin-resistant gastric cancers (45). These results indicated that specific bacteria may cause the abnormal methylation of the host genes and then silence the expression of the genes, which may contribute to the distal metastasis of cancer cells. In addition, the methylated features tended to further promote the distal metastasis of STAD by affecting the richness or evenness of intratumoral bacteria. Survival mediation analyses indicated that *K. sedentarius* and *A. oris* interact strongly with methylation changes of immune genes related to prognosis. There was a mutual mediation effect between *K. sedentarius* and the hypermethylation of the cg26279814 site of the *CDK12* gene. Previous studies have shown that the expression level of *CDK12* in gastric cancer was correlated with advanced stages and poor outcomes and was positively correlated with the CD8 cell density (46, 47). The potential of *CDK12/PAK2* as a therapeutic target for patients with gastric cancer was highlighted in a previous study (48). There was a mutual mediation effect between *Streptococcus sanguinis* and the hypermethylation of the cg12570942 site of the *DTYMK* gene, which is a potential biomarker for hepatocellular carcinoma (49, 50). In summary, the bi-directional mediation effect between methylation changes and microbial characteristics may promote distal metastasis and affect the prognosis of STAD patients.

The tumor microbiota-host epigenetic axis provides a new perspective on the mechanism of cancer development and potential clinical intervention targets. Bacterial cancer therapies have been investigated (51). The deletion of some virulence genes of bacteria such as *S. saccharolyticus*, *K. sedentarius*, and *A. oris* that are closely related to host epigenetic changes could be targeted to related genes in tumor cells to regulate the growth of cancer cells. Epigenetic therapies can also be targeted to specific genomic loci to effectively activate or silence specific genes such as *ZNF215*, *PTGS2*, *CDK12*, and *DTYMK* (52). Since our research results were limited to correlative data, and some of them were not casual, further *in vitro* and *in vivo* experiments and longitudinal cohorts are necessary to confirm our calculation results. Diet, medications, age, and race are important factors that contribute to host DNA methylation changes and also affect the microbiome, which should not be ignored and may affect our results.

## MATERIALS AND METHODS

**Data acquisition.** For patients with STAD, the TCGA clinical data ($n = 407$), RNA sequencing data ($n = 407$) and Illumina Human Methylation 450 BeadChip data ($n = 368$) were downloaded from UCSC XENA (http://xena.ucsc.edu/), and the raw BAM files of the RNA sequencing data were obtained from the NCI Genomic Data Commons. The clinical information of the patients is shown in Table 1.

**Metagenomic profiling based on RNA sequencing data for STAD.** Based on mapping information in the raw BAM files of the RNA sequencing data, reads that did not align with the known human reference genome (hg38) were extracted using samtools (v1.9) and bedtools (v2.30.0) and filtered using Trimmomatic (v0.39). To obtain the microbial reads and their taxonomic annotations, the retained reads were aligned against the NCBI nonredundant database using Kraken 2 and Bracken, which produced species- and genus-level abundance estimates (k-mers = 35), and the reads that could not be aligned to bacteria were filtered out (53). Functional annotation and classification of proteins were performed by using eggNOG-Mapper v2 (54), PICRUSt, and Bakta analysis (55).

**TABLE 1** Demographic characteristics of patients with and without distal metastasis of cancer cells[a]

| Characteristic | Value for clinical M stage group | | P value |
| --- | --- | --- | --- |
| | M0/MX (n = 319) | M1 (n = 19) | |
| Age (yrs) | 65.8 (10.4) | 58.1 (10.7) | |
| | | | |
| No. of patients of sex | | | 0.668 |
| Male | 209 | 11 | |
| Female | 110 | 8 | |
| | | | |
| No. of patients at clinical T stage | | | 0.008 |
| T1 | 18 | 0 | |
| T2 | 67 | 0 | |
| T3 | 152 | 8 | |
| T4 | 82 | 11 | |
| | | | |
| No. of patients at clinical N stage | | | 0.101 |
| N0 | 101 | 2 | |
| N1 | 82 | 5 | |
| N2 | 67 | 4 | |
| N3 | 61 | 8 | |
| NX | 8 | 0 | |
| | | | |
| No. of patients with survival outcome | | | 0.681 |
| Alive | 186 | 10 | |
| Dead | 123 | 9 | |
| | | | |
| Survival time (days) | 630 (544) | 513 (589) | 0.422 |
| | | | |
| No. of patients with disease types | | | 0.999 |
| Adenomas and adenocarcinomas | 289 | 17 | |
| Cystic, mucinous, and serous neoplasms | 30 | 2 | |

[a]The numbers in parentheses indicated the sample size. Primary tumor (T), regional lymph nodes (N).

**Contamination control and evaluation.** Analysis of any batch effect arising due to sequencing was conducted using the decontam R package with default settings (56). We filtered the possible contaminants in different sequencing centrals and plates and removed genera typically found in "negative blank" reagents (n = 92). Next, we reallowed the genera that were related to the "stomach neoplasm" phenotype in GMrepo v2, which uses a standardized process to reanalyze microbial metagenome and 16S rRNA gene sequencing data and allows users to find bacteria related to phenotypes (57). In general, we filtered out 68 genus-level and 593 species-level taxa.

**Intratumoral microbiome analysis of STAD.** To analyze the intratumoral microbiome profile of STAD, we compared the microbiota diversity and taxa of the intratumoral microbiota between STAD and adjacent control tissues as well as between STAD tissues with and without metastasis. In brief, the phyloseq package (v1.34.0) was used to analyze the alpha diversity at the species level measured by the observed, Shannon, and Simpson indices; principal-coordinate analysis was used to analyze beta diversity (Bray-Curtis distance) using the vegan package (v2.5.7); and the LEfSe algorithm, which compares the relative abundances of all bacterial taxa and estimates the influence of the abundance of each component (species) on the difference effect, was used to select the bacteria with significant differences in relative abundance between groups.

Next, Boruta-RandomForest (Boruta package and RandomForest packages, 5-fold cross-validation, 70% samples were randomly divided into training set) analysis was employed to select STAD occurrence-associated microbe features among differentially abundant bacteria (58). This method ranks the importance of all features and finally screens out the features most related to the grouping through multiple permutations. The AUC and the AUPR curve were obtained to assess the model's performance on a validation set (ROCR package).

Next, we aimed to explore whether the intratumoral microbiota and the microbiota functional genetic makeup could predict or affect the survival of STAD patients. The microbiota data were input into the model after library size correction using logcpm and snm (Seurat and snm packages). We first used univariate Cox analysis to identify potential microbe features, functional genes, and pathways. By using the glment package, LASSO-penalized Cox regression analysis, which realizes variable selection and solves multicollinearity problems for survival data, was then conducted to further select likely prognostic microbes for OS in patients. After STAD patients were randomly separated into a training set and a test set (7:3), potential prognostic microbes were included in the multivariate Cox regression model, and the predictive performances for the 1-, 3-, and 5-year survival rates of patients were evaluated on the test set. The risk scores were calculated and compared among the different survival status groups. MetagenoNets (https://web.rniapps.net/metagenonets/), an integrated and visualized platform for existing network analysis methods, was employed for cooccurrence network

analysis based on compositional data through Lasso (CCLasso) correlation algorithms at the genus level and identified hub genera separately ($r > 0.8$; $P < 0.01$).

**DNA methylation analysis of STAD.** Before the analysis, the methylation data were quality controlled and normalized by using the CHAMP package, which is an integrated package for analyzing 450k methylation for Illumina data (59). To explore whether the DNA methylation pattern is involved in the occurrence and distal metastasis of STAD, the DMPs and DMRs between the M0/MX and the M1 groups were determined using the CHAMP package. The methylated genes were clustered to identify interactome hot spots of differential promoter methylation. KEGG analysis of differentially methylated genes was conducted using the Database for Annotation Visualization and Integrated Discovery (DAVID), which is a comprehensive set of functional annotation tools (https://david.ncifcrf.gov/) (60). Univariate Cox regression analysis was performed for OS to identify DMPs and DMRs as potential methylation changes (adjusting for gender and age [FDR of <0.25]) and also to identify DEGs as DNA methylation has a substantial impact on gene expression. LASSO-Cox regression analysis was then applied to construct a prognostic model to identify the relevant prognostic methylation changes based on the potential DMGs or DMRs. Similar to the microbiome analysis, multivariate Cox regression was used to tune the model, and the performance of the model was evaluated. The risk scores were calculated and compared among the different survival status groups. The Spearman correlation coefficient was introduced to measure the relationship between gene methylation changes and expression levels.

**Bi-directional mediation analysis.** The approach for the potential causal mediation analysis was based on a counterfactual framework in which the total effect of $X$ to $Y$ can be decomposed into the direct effect of $X$ to $Y$ and the indirect effect of $X$ to $Y$ through the mediation variable $M$. In this study, $X$ and $M$ can be both microbial characteristics and methylation changes. $Y$ can be whether STAD shows distal metastasis or survival.

Survival mediation analyses, which determine the special direct and indirect effects for the survival data, were performed by employing the timereg package (v2.0.1) to investigate the predictive ability of the microbial features selected by LASSO-Cox analysis or the methylated features identified by univariate Cox Proportional Hazards (CoxPH) analysis for the survival of STAD patients that could be explained by other type features (61).

We then identified the underlying causal role of the microbiota in contributing to the distal metastasis of STAD through methylation changes, microbial features (microbial diversity indices and significant differentially abundant taxa between the M0/MX and M1 groups), and methylated features (including DMRs and hot spot modules) as candidate $X$ or $M$ (Mediation package) (62). The specific steps are available in the supplemental material. Finally, notable microbe and methylation change mediation interaction networks were constructed from statistically significant potential microbe-methylation casual interactions imported into Cytoscape 3.8.0 (63).

**Cell lines and culture of *Staphylococcus saccharolyticus*.** The HGC-27 human gastric cancer tissue cell line, the AGS human gastric cancer tissue cell line, and the GES-1 human normal gastric epithelial cell line were purchased from the ATCC. The HGC-27 cell line was cultured in RPMI 1640 medium supplemented with 20% fetal bovine serum (FBS) and 1% penicillin-streptomycin. The AGS cell line was cultured in F-12K medium supplemented with 10% FBS and 1% penicillin-streptomycin. The GES-1 cell line was cultured in Dulbecco's modified Eagle's medium (DMEM) supplemented with 10% FBS and 1% penicillin-streptomycin. All cells were cultured at 37°C in a humidified incubator with 5% $CO_2$.

*S. saccharolyticus* was purchased from the Japan Collection of Microorganisms (JCM) (catalog number bio-123834). The bacterial liquid suspension was cultured in liquid culture medium (nutrient broth) and enriched at 37°C for 36 h under aerobic conditions. After measuring the optical density at 600 nm ($OD_{600}$) and counting bacterial CFU, a bacterial standard curve was obtained to calculate the bacterial fluid volume needed for coculture experiments.

**Design of cell experiments.** There were 6 groups in the cell experiments (see Fig. S5 in the supplemental material). In group 1, the cell lines were cocultured with *S. saccharolyticus* for 6 h. In group 2, the cell lines were cocultured with *S. saccharolyticus* for 6 h and then transfected with a plasmid overexpressing the *ZNF215* gene [pcDNA3.1(+) overexpressing the *ZNF215* gene]. In group 3, the cell lines were cocultured with *S. saccharolyticus* for 6 h and then transfected with a negative-control plasmid with an overexpressed gene [pcDNA3.1(+)]. In group 4, cell lines were transfected with a plasmid silencing the *ZNF215* gene (pLVX-shRNA1 plus knockdown of the *ZNF215* gene). In group 5, the cell lines were transfected with a negative-control plasmid with a silent gene (pLVX-shRNA1). Group 6 was the blank control group for the cell lines. The multiplicity of infection (MOI) was 10:1 in these groups.

**CCK8 cell proliferation experiment.** Cell counting kit 8 (CCK8) analysis was carried out to determine the regulation of cell proliferation. Cells were seeded into 96-well plates at a density of 1,000 cells per well with 100 $\mu$L of cell culture medium. At 0 h, 24 h, and 48 h of transfection, a CCK8 solution diluted in phosphate-buffered saline (PBS) at 9:1 was added, and the absorbance at 450 nm was measured 1 h later.

**Cell clone formation experiment.** Cells were seeded into a 6-well plate (2,000 cells/well). The cultures were maintained at 37°C in a 5% $CO_2$ incubator for 5 days. The colonies were fixed with 4% paraformaldehyde and stained with 0.5% crystal violet. The numbers of colonies were photographed and counted under a microscope to evaluate the proliferation ability of the cells.

**RNA extraction and quantitative real-time PCR analysis.** Total mRNA from the cell lines was extracted with FastPure cell/tissue total RNA isolation kit V2 (Vazyme) within 48 h and then reverse transcribed using the HiScript III RT supermix for qPCR (+gDNA wiper) (Vazyme). The qPCR analyses were performed using ChamQ universal SYBR qPCR master mix (Vazyme). mRNA expression was assessed by the comparative cycle threshold ($C_T$) method ($2^{-\Delta\Delta C_T}$); human beta-actin was used as a housekeeping gene.

**Statistical analyses.** All statistical analyses were done and graphs were created using R software (version 4.0.3). A Wilcoxon test was used for comparisons between two groups. Permutational multivariate analysis of variance was used to determine beta diversity by using the vegan package (v2.5.7). Spearman rank tests were conducted to analyze the correlations between key microbes and methylation changes. A survival curve was estimated using a Kaplan-Meier curve, and differences were tested using the log rank test. The Benjamini-Hochberg method (64) was used to correct for multiple testing. *P* values of <0.05 were considered to indicate statistically significant differences unless indicated otherwise. The Venn diagram was visualized using Evenn Web (http://www.ehbio.com/test/venn/).

**Data availability.** The raw RNA sequencing data sets from TCGA used and analyzed in the current study are available from the corresponding author upon reasonable request. The data generated in this study are publicly available in the supplemental material. The scripts used to access, manage, and run the data can be found at our GitHub repository at https://github.com/MicrobiomeXsdu/TCGA_STAD.git.

## SUPPLEMENTAL MATERIAL

Supplemental material is available online only.
**SUPPLEMENTAL FILE 1**, DOCX file, 7.3 MB.
**SUPPLEMENTAL FILE 2**, XLSX file, 7 MB.
**SUPPLEMENTAL FILE 3**, XLSX file, 5.7 MB.
**SUPPLEMENTAL FILE 4**, XLSX file, 0.04 MB.

## ACKNOWLEDGMENTS

This study was funded by National Natural Science Foundation of China grant 82172320, TaiShan Industrial Experts Program grant tscy20190612, and the Shandong University Outstanding Young Scholars Program.

We declare no potential conflicts of interest.

Conceptualization, Kaile Yue, Dashuang Sheng, Xinxin Xue, and Lei Zhang; Data Curation, Kaile Yue, Dashuang Sheng, and Xinxin Xue; Funding Acquisition, Lei Zhang and Guoping Zhao; Investigation, Kaile Yue, Dashuang Sheng, and Xinxin Xue; Methodology, Kaile Yue, Dashuang Sheng, Xinxin Xue, and Chuandi Jin; Project Administration, Kaile Yue, Dashuang Sheng, and Xinxin Xue; Resources, Guoping Zhao, Chuandi Jin, and Lei Zhang; Supervision, Guoping Zhao, Chuandi Jin, and Lei Zhang; Validation, Lanlan Zhao; Visualization, Kaile Yue, Dashuang Sheng, Xinxin Xue, and Lanlan Zhao; Writing – Original Draft, Kaile Yue, Dashuang Sheng, Xinxin Xue, and Lanlan Zhao; and Writing – Review & Editing, Guoping Zhao, Chuandi Jin, and Lei Zhang.

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
