## [Reviewer comments · Microbiology Spectrum]

Microbiology Spectrum

Bi-directional mediation effects between intratumoral microbiome and host DNA methylation changes contribute to Stomach Adenocarcinoma

Kaile Yue, Dashuang Sheng, Xinxin Xue, Lanlan Zhao, Guoping Zhao, Chuandi Jin, and Lei Zhang

Corresponding Author(s): Lei Zhang and Chuandi Jin, Shandong University

Review Timeline:

Submission Date:	March 1, 2023
Editorial Decision:	March 21, 2023
Revision Received:	April 29, 2023
Accepted:	May 6, 2023

Editor: Zhenjiang Xu

Reviewer(s): Disclosure of reviewer identity is with reference to reviewer comments included in decision letter(s). The following individuals involved in review of your submission have agreed to reveal their identity: Tingtao Chen (Reviewer #1); Hui-Xin Liu (Reviewer #2)

Transaction Report:

DOI: <https://doi.org/10.1128/spectrum.00904-23>

March 21, 2023

Prof. Lei Zhang
Shandong University
Department of Biostatistics
Jinan
China

Re: Spectrum00904-23 (Bi-directional mediation effects between intratumoral microbiome and host DNA methylation changes contribute to Stomach Adenocarcinoma)

Dear Prof. Lei Zhang:

Link Not Available

Sincerely,

Zhenjiang Xu

Journals Department
Reviewer comments:

Reviewer #1 (Comments for the Author):

Based on RNA-seq and DNA methylation data, Yue et al. studied the influence of intratumoral microbiome and host DNA methylation on metastasis and prognosis of gastric cancer. An in vitro experiment was further conducted to verify their findings. The study of intratumoral microbiome is a fast-growing field of investigation and is so important to unravel the mechanisms by which specific bacteria taxa and gene are associated with cancer progress and control. The study design is good, and I have several minor questions and suggestions.

Comments:

1. The study used RNA-seq data to analyze the microbial composition within gastric cancer tissues. I wonder how many of the

- non-human sequences were matched to microbes. It is recommended that the author add this section to the article.
2. It is mentioned in the results section that A total of 20 potential mediation linkages related to distal metastasis of STAD were revealed by bi-directional mediation analysis. Why ZNF215 gene and *S. saccharolyticus* were selected to further verified by cell lines? What is the selection reason?
 3. In this study, microbial and methylation data were used to construct survival prognostic models for gastric cancer. And the 95% confidence interval of these models were showed. What is the C-index of these models?
 4. The functional genes of bacteria were also analyzed, and several significant and meaningful microbial genes were screened out, such as *fliZ* gene and *YafN* gene, but this part was not discussed enough in this paper. Please add it in the discussion section.
 5. The authors compared the microbial profiles called by themselves with those called by Poore G et al. to strengthen the validity of their results. What's the development are made in this paper compared with the methods used in Poore G et al.
 6. Kraken 2 was the used to alignment in this study. What is the advantage of this method compared with Kraken? And please clarify the purpose of using both Kraken 2 and Bracken.
 7. Please elaborate on the samples selected for your different omics study. And clarification is required for overlapping patient information for both RNAseq data and DNA Methylation data. Table with detailed information of samples used in this study would be useful.
 8. The manuscript mentioned "A total of 10.8% RNA sequencing reads of The Cancer Genome Atlas (TCGA) STAD were defined as non-human reads". Were those reads have any batch effect arising due to sequencing? The methods of quality control and batch correction in the process of obtaining microbiota data should be more detailed.
 9. What is the definition of the interaction hotspot of methylation in manuscript? How did the authors build those networks?

Reviewer #2 (Comments for the Author):

The authors exploited RNA-seq data and DNA methylation data of stomach adenocarcinoma from TCGA to investigate the bi-directional mediation effect between intratumoral bacteria and host DNA methylation with distal metastasis and prognosis of patients. The data sources from the existing cancer-related databases were well used, and the experimental verification proved that the methylation change of host gene from bacteria to further affects the growth of cells in this study. This research design meets the requirements of journal. However, some minor issues still need to be improved.

1. There were innumerable grammatical errors consistent with non-native speakers. The authors should solicit the aid of a native or fluent English speaker to edit the text and check spelling mistakes. In the results, the authors made some bold statements about prognostic markers identification in STAD. Since this was a relevant study, the author should be cautious when making such a statement.
2. The idea is not novel because earlier studies have utilized RNA seq data. Compared with previous studies using RNA-seq data, what were the standard process for obtaining microbial sequences from RNA-seq sequencing data and the innovations of this study? What were the proportions of sequencing reads were classified as non-human and then were assigned to bacteria? What efforts were made to control possible sequencing contaminations?
3. The methods for this paper need minor revisions. Several tools and packages were used for microbiome and methylation analysis. However, the rationale and advantages of using those tools should be explained.
4. The detailed criteria to select patients for microbiome, methylation and interaction analysis should be list. If possible, the author could supplement the host DNA methylation analysis grouped by sample tissue type, which provide clues for how the presence of bacterial species affects methylation and gene expression, and then induce carcinogenesis.
5. The in vitro cell experiment provided by the author was reliable. However, the basis for the selection of verification results was not clearly indicated. Please explain the reasons for verifying the mediation effect between *Staphylococcus saccharolyticus* and ZNF215 gene.

Staff Comments:

Preparing Revision Guidelines

To submit your modified manuscript, log onto the eJP submission site at <https://spectrum.msubmit.net/cgi-bin/main.plex>. Go to Author Tasks and click the appropriate manuscript title to begin the revision process. The information that you entered when you first submitted the paper will be displayed. Please update the information as necessary. Here are a few examples of required

updates that authors must address:

Please return the manuscript within 60 days; if you cannot complete the modification within this time period, please contact me. If you do not wish to modify the manuscript and prefer to submit it to another journal, please notify me of your decision immediately so that the manuscript may be formally withdrawn from consideration by Microbiology Spectrum.

Dear Editors,

On behalf of my co-authors, we thank you for giving us a chance to express our opinion and improve the quality of our article. We have read your and reviewers' comments carefully and tried our best to revise our manuscript according to the comments. Modified places were marked in blue. Attached is the revised version, which we would like to submit for your kind consideration. Here, we would like to explain the changes as follows:

In response to the question raised by the Reviewer #1.

Based on RNA-seq and DNA methylation data, Yue et al. studied the influence of intratumoral microbiome and host DNA methylation on metastasis and prognosis of gastric cancer. An in vitro experiment was further conducted to verify their findings. The study of intratumoral microbiome is a fast-growing field of investigation and is so important to unravel the mechanisms by which specific bacteria taxa and gene are associated with cancer progress and control. The study design is good, and I have several minor questions and suggestions.

QUESTION 1. The study used RNA-seq data to analyze the microbial composition within gastric cancer tissues. I wonder how many of the non-human sequences were matched to microbes. It is recommended that the author add this section to the article.

ANSWER: Thanks to the reviewer for helpful comments.

From the 10.8% non-human RNA sequencing reads, 18.8% of non-human reads after quality control had been mapped in Kraken 2.

We have added this sentence to the results section, "A total of 10.8% RNA sequencing reads of The Cancer Genome Atlas (TCGA) STAD were defined as non-human reads, 18.8% of non-human reads after quality control had been mapped in Kraken 2 after decontamination and 1,236 genus and 4,597 species were identified, respectively." (Page 9, Line 117 to 120).

QUESTION 2. It is mentioned in the results section that a total of 20 potential mediation linkages related to distal metastasis of STAD were revealed by bi-directional mediation analysis. Why *ZNF215* gene and *S. saccharolyticus* were selected to further verified by cell lines? What is the selection reason?

ANSWER: Thanks for raising this important issue.

The reasons why we chose to verify the bioinformatics analysis results of *ZNF215* gene and *S. saccharolyticus* were as follows:

(1) Metastasis accounts in part for the high mortality from gastric cancer(2). Using CCK-8 cell proliferation experiment and cell clone formation experiment could well reflect the distant invasion and cloning ability of cells, which was the basis of distant metastasis of cancer cells. Therefore, we chose the results of bi-directional mediation analysis results related to distant metastasis of cancer cells for verification.

(2) The bi-directional mediation analysis revealed that three of the twenty potential mediation linkages related to distant metastasis of STAD were related to abnormal methylated genes and bacteria. The results of data analysis showed that the bi-directional mediation effect between *ZNF215* gene and *S. saccharolyticus* would affect the distant

metastasis of cancer cells. In the other two potential mediation linkages, it is suggested that bacteria were the mediations of epigenetic changes from host to distant metastasis of cancer cells. At present, most studies tended to regard bacteria as the cause of abnormal genetic changes(3-5). So, we used cell experiments to determine the direction of effect between them.

(3) The selection of *ZNF215* gene and *S. saccharolyticus* could provide clues for clinical treatment. *Staphylococcus* was also the dominant bacteria of stomach microbiota, and it was urease-producing organisms. Now, it has been identified that some species of *Staphylococcus* were prevalent in patients with dyspepsia. LEfSe results showed that the relative abundance of *S. saccharolyticus* increased in patients with metastasis. By deleting some virulence genes of *S. saccharolyticus* could be targeted to the related genes in tumor cells to regulate the growth of cancer cells. And previous studies have shown that the *ZNF215* genes can be used as a diagnostic and prognostic biomarker for the basal-like breast cancer and acute myeloid leukemia(6, 7). Differential analysis of host DNA methylation showed that specific regions of *ZNF215* gene in metastatic group were hypermethylated and negatively correlated with gene expression level. Epigenetic therapies can also be targeted to specific genomic loci to effectively activate *ZNF215* gene.

QUESTION 3. In this study, microbial and methylation data were used to construct survival prognostic models for gastric cancer. And the 95% confidence interval of these models were showed. What is the C-index of these models?

ANSWER: Thank you for your nice suggestions.

We calculated the performance of microbes, DMPs and DMRs screened by lasso-cox on the validation set, and the C index was 0.622, 0.612 and 0.710 by using glmnet package respectively.

QUESTION 4. The functional genes of bacteria were also analyzed, and several significant and meaningful microbial genes were screened out, such as *fliZ* gene and *YafN* gene, but this part was not discussed enough in this paper. Please add it in the discussion section.

ANSWER: We are grateful for the suggestion.

As suggested by the reviewer, we have added more details about the functional genes of bacteria in the discussion section. The relevant discussions were amended as follows:

(1) "The expression of *fliZ* gene of bacteria increased significantly in cancer tissues, especially in patients with distant metastasis of cancer cells. Previous studies have shown that *fliZ* gene can regulate the pathogenicity of bacteria." (Page 15, Line 292 to 295).

(2) "*YafN*, a microbial functional gene encoding antitoxin-toxin operon, was not only related to the survival of patients, but also related to whether cancer cells metastasize." (Page 15, Line 325 to 327).

QUESTION 5. The authors compared the microbial profiles called by themselves with those called by Poore G et al. to strengthen the validity of their results. What's the development are made in this paper compared with the methods used in Poore G et al.

ANSWER: We appreciate you raising these important points.

The paper by Poore G et al. (Nature 2020) utilized Kraken to evaluate the microbial community in TCGA tissue samples, which given us great inspiration and was the focus of our article. Compared with Poore G et al., this paper made improvements and further explorations. The developments are made in this paper compared with the methods used in Poore G et al as follow:

(1) We carried a more detailed classification of bacteria and annotate the microbial sequence to the species level. In our manuscript, Kraken 2 and Bracken were used to obtain the microbial reads and their taxonomic annotations.

(2) We added the quality control of sequences and carried out optimized steps to decontaminated.

1) Specifically, we measured the quality of those reads by using FastQC (v0.11.8) and MultiQC (v1.9), and then filtered the poor quality reads by using Trimmomatic (v0.39) with the following options: ILLUMINACLIP:TruSeq2-PE.fa:2:30:10 SLIDINGWINDOW:4:20 MINLEN:35.

2) We re-allowed the genera related to the “Stomach Neoplasms” phenotype in GMrepo, which used a standardized process to re-analyze 253 microbial metagenome and 16S rRNA gene sequencing data and allowed users to find bacteria related to phenotypes.

(3) We not only paid attention to the composition of microbiome, but also paid attention to the functional gene composition of bacteria and its enrichment KEGG pathway.

(4) We correlated microbial data with host DNA methylation level, which provided clues for microbial host interaction mechanism.

QUESTION 6. Kraken 2 was the used to alignment in this study. What is the advantage of this method compared with Kraken? And please clarify the purpose of using both Kraken 2 and Bracken.

ANSWER: We thank for the thoughtful suggestion.

(1) Kraken 2 provides a fast-taxonomic classification of sequence data and improves upon Kraken 1 by reducing memory usage by 85%, allowing greater amounts of reference genomic data to be used.

(2) Generally speaking, Kraken and Bracken were used to produce accurate species- and genus-level abundance estimates in this manuscript.

(3) Bracken is a highly accurate statistical method that computes the abundance of species in DNA sequences from a metagenomics sample. Bracken uses the taxonomy labels assigned by Kraken, a highly accurate metagenomics classification algorithm, to estimate the number of reads originating from each species present in a sample. Kraken classifies reads to the best matching location in the taxonomic tree, but does not estimate abundances of species. We use the Kraken database itself to derive probabilities that describe how much sequence from each genome is identical to other genomes in the database, and combine this information with the assignments for a particular sample to estimate abundance at the species level, the genus level, or above.

QUESTION 7. Please elaborate on the samples selected for your different omics study. And clarification is required for overlapping patient information for both RNAseq data and DNA Methylation data. Table with detailed information of samples used in this study would be useful.

ANSWER: Thank you for pointing this out.

The data sources of this study from the 338 cancer tissue samples of 375 patients and 69 matched normal tissue samples adjacent to cancer with RNA-seq data, and the cancer tissue samples of 368 patients with HumanMethylation450 data.

(1) A total of 338 patients had RNA sequencing data and DNA methylation sequencing data, of which, 19 patients with clinical M stage of M1. The specific clinical information was shown in Table 1.

(2) In the microbial analysis, the analysis grouped by cancer and adjacent tissue types is based on 338 cancer tissue samples of 375 patients and 69 matched normal tissue samples adjacent to cancer. Survival analysis, analysis of group by whether patients were metastatic and mediation analysis of microbiome and DNA methylation were based on the data of 338 patients with both RNA-seq data and HumanMethylation450 data. To facilitate understanding, we summarized it in **Response table 1**.

Response table 1. The sample number used in different omics analysis

Omics	Tumor vs. adjacent control tissues	M0/MX vs. M1	Survival outcome and time
Microbiome analysis	407	338	338
DNA methylation analysis	-	338	338
Bi-mediation analysis	-	338	338

QUESTION 8. The manuscript mentioned "A total of 10.8% RNA sequencing reads of The Cancer Genome Atlas (TCGA) STAD were defined as non-human reads". Were those reads have any batch effect arising due to sequencing? The methods of quality control and batch correction in the process of obtaining microbiota data should be more detailed.

ANSWER: Thanks for raising this important issue.

(1) We used the following methods to reduce the batch effect that may be caused by sequencing.

1) As to the batch effect arising due to sequencing in different centers or plates may be exit, we used the decontam R package by using default $P^*=0.1$ hyperparameter value for decontam for both "prevalence" (i.e. blank-based) and "frequency" (i.e. concentration-based) modes of decontamination to measure likely contaminations. This step identified 28 genera and 141 species as likely contaminations.

2) Then the microbes in the "negative" blank controls list (70 bacteria) of contaminations which included the bacteria in DNA extraction blanks, DNA library preparation blanks and empty control wells were removed.

3) We re-allowed the genera related to the "Stomach Neoplasms" phenotype in GMrepo. In general, 68 genera and 593 species-level taxa were filtered as possible pollutants,

which reduced the batch effect arising by sequencing.

4) We also adjusted library size of microbiota sequencing by logcpm and snm (Seurat and snm packages).

(2) For the obtain of microbial data, we adopt the following processes:

1) Reads that did not align to known human reference genomes (hg38) was extracted using samtools (v1.9) and bedtools.

2) We measured the quality of those reads by using FastQC (v0.11.8) and MultiQC (v1.9), and then filtered the poor quality reads by using Trimmomatic (v0.39) with the following options: ILLUMINACLIP:TruSeq2-PE.fa:2:30:10 SLIDINGWINDOW:4:20 MINLEN:35.

3) To obtain the microbial reads and their taxonomic annotations, these retained reads were aligned against the NCBI non-redundant database using Kraken 2 and Bracken.

4) The quality of microbiota reads was measured by using FastQC (v0.11.8) and MultiQC (v1.9).

5) Contamination control and evaluation. We use the decontam R package, "negative" blank controls list and the genera related to "Stomach Neoplasms" phenotype in GMrepo to decontaminated

6) Function annotation and classification by eggNOG-Mapper and PICRUSt.

This process was optimized according to Poore G et al(8). We added quality control by FastQC (v0.11.8) and MultiQC (v1.9), upgrade the bacterial reference database to Kraken 2, and added microbial functional gene annotation by eggNOG-Mapper and PICRUSt.

QUESTION 9. What is the definition of the interaction hotspot of methylation in manuscript? How did the authors build those networks?

ANSWER: Thank you for your comment.

The champ.EpiMod function in CHAMP package was used to identify interactome hotspots of differential promoter methylation(9). By "interactome hotspot" we mean a connected subnetwork of the protein interaction network (PIN) with an exceptionally large average edge-weight density in relation to the rest of the network. We call these "hotspots" also Functional Epigenetic Modules (FEMs). The weight edges are constructed from the statistics of association of DNA methylation with the phenotype of interest. Thus, the EpiMod algorithm can be viewed as a functional supervised algorithm, which uses a network of relations between genes, to identify subnetworks where a significant number of genes are associated with a phenotype of interest.

-----End of Reply to Reviewer #1-----

In response to the question raised by the Reviewer #2.

The authors exploited RNA-seq data and DNA methylation data of stomach adenocarcinoma from TCGA to investigate the bi-directional mediation effect between intratumoral bacteria and host DNA methylation with distal metastasis and prognosis of patients. The data sources from the existing cancer-related databases were well used, and the experimental verification proved that the methylation change of host gene from bacteria to further affects the growth of cells in this study. This research design meets the

requirements of journal. However, some minor issues still need to be improved.

QUESTION 1. There were innumerable grammatical errors consistent with non-native speakers. The authors should solicit the aid of a native or fluent English speaker to edit the text and check spelling mistakes. In the results, the authors made some bold statements about prognostic markers identification in STAD. Since this was a relevant study, the author should be cautious when making such a statement.

ANSWER: We appreciate you raising these important points.

(1) According to your important suggestion, we corrected the grammatical and spelling errors and polish the whole manuscript by native speakers and experts, specially sentence structure.

(2) We revised the corresponding descriptions in the results section as follows:

1) "The univariant Cox analysis identified 278 species as potential prognostic factors for overall survival (OS) (Table S8)" (Page 8, Line 155 to Page 9, Line 156)

2) "Taken together, intratumoural microbiota were found to involve in the occurrence, distal metastasis and prognosis of STAD which shown likely diagnostic and prognostic value as well." (Page 9, Line 166 to 168)

3) "The Venn diagram showed that both methylation and expression of 10 genes have potential prognostic value for patients with STAD." (Page 10, Line 191 to 193)

4) "Generally speaking, STAD patients with distant metastasis may be related with more hypermethylation genes and some aberrant methylation sites and regions have potential predictive ability for prognosis." (Page 10, Line 194 to 196)

5) "There was a potentially bidirectional mediating effect between the abundance change of *S. saccharolyticus* and the methylated change of *ZNF215* gene, and *S. saccharolyticus* also mediated the effect of methylated change of *PTGS2* gene on metastasis." (Page 11, Line 205 to 208)

6) "For instance, *K. sedentarius* may contribute to the prognosis of STAD patients by affecting nine methylation features (34.9%), meanwhile, it could also mediate the effect of these nine methylation characteristics on the prognosis (45.1%). *A. oris* could contribute to the prognosis of STAD patients by affecting 6 methylation features (23.3%), meanwhile, it also mediated the effect of 2 of these 6 methylation features and the other 2 methylation features on the prognosis (34.2%). There may be a bi-mediating effect between *Streptococcus sanguinis* and the hypermethylated of the cg12570942 sites of *DTYMK* gene." (Page 11, Line 214 to Page 12, Line 222)

QUESTION 2. The idea is not novel because earlier studies have utilized RNA seq data. Compared with previous studies using RNA-seq data, what were the standard process for obtaining microbial sequences from RNA-seq sequencing data and the innovations of this study? What were the proportions of sequencing reads were classified as non-human and then were assigned to bacteria? What efforts were made to control possible sequencing contaminations?

ANSWER: Thank you for pointing this out.

(1) The standard process for obtaining microbial sequences from RNA-seq data was as follows.

- 1) Extract non-human reads and quality control.
 - 2) Obtain the microbial reads and their taxonomic annotations and quality control.
 - 3) Microbiota function annotation and classification.
 - 4) Contamination control.
 - 5) Downstream bioinformatics analysis.
- (2) Compared with Poore G et al., this paper made improvements and further explorations.
- 1) We carried Kraken 2 to obtain detailed classification of bacteria and annotate the microbial sequence to the species level
 - 2) We added the quality control of sequences to filter the lower quality reads.
 - 3) We described a comprehensive microbial map of gastric cancer solid tissue, including species composition, microbial functional gene composition and KEGG functional pathway enriched.
 - (3) From the 10.8% non-human RNA sequencing reads, 18.8% of non-human reads after quality control had been mapped to reference database of bacteria by using Kraken 2.
- We have added this sentence to the results section, “A total of 10.8% RNA sequencing reads of The Cancer Genome Atlas (TCGA) STAD were defined as non-human reads, 18.8% of non-human reads after quality control had been mapped in Kraken 2 after decontamination and 1,236 genus and 4,597 species were identified, respectively.” (Page 9, Line 117 to 120).
- (4) We controlled the batch effect that may be caused by sequencing by filtering the quality of sequencing reads and removing possible contaminations.
 - 1) We measured the quality of non-human reads by using FastQC and MultiQC, and then filtered the poor-quality reads (Remove the Primer Adapter and the sliding window with an average quality below 20) by using Trimmomatic.
 - 2) As to the batch effect arising due to sequencing in different centers or plates may be exit, we used default parameters of decontam R package to measure likely contaminations. This step identified 28 genera and 141 species as likely contaminations.
 - 3) Then the microbes in the “negative” blank controls list (70 bacteria) of contaminations which included the bacteria in DNA extraction blanks, DNA library preparation blanks and empty control wells were further removed.
 - 4) Finally, we re-allowed the genera related to the “Stomach Neoplasms” phenotype in GMrepo web, which allowed users to find bacteria related to specific phenotypes. In general, 68 genera and 593 species-level taxa were filtered as possible pollutants, which reduced the batch effect arising by sequencing.
 - 5) The microbiota data were input into the model after library size correction by logcpm and snm (Seurat and snm packages) to reduce the influence of different sequencing depths on data analysis results.

QUESTION 3. The methods for this paper need minor revisions. Several tools and packages were used for microbiome and methylation analysis. However, the rationale and advantages of using those tools should be explained.

ANSWER: We gratefully appreciate for your valuable suggestion. We supplemented the advantages and explanations of key methods and software and cited relevant references.

(1) "Next, we re-allowed the genera which were related to the "Stomach Neoplasms" phenotype in GMrepo V2, which used a standardized process to re-analyze microbial metagenome and 16S rRNA gene sequencing data and allowed users to find bacteria related to phenotypes." (Page 20, Line 400 to 403)

(2) "LEfSe which compared the relative abundance of all bacterial taxa and estimated the influence of each component (species) abundance on the difference effect was used to select the bacteria with significant differences in relative abundance between groups." (Page 20, Line 412 to 414)

(3) "The KEGG analysis of differential methylated genes was conducted using Database for Annotation Visualization and Integrated Discovery (DAVID) which is a comprehensive set of functional annotation tools (<https://david.ncifcrf.gov/>)." (Page 21, Line 443 to 446)

QUESTION 4. The detailed criteria to select patients for microbiome, methylation and interaction analysis should be list. If possible, the author could supplement the host DNA methylation analysis grouped by sample tissue type, which provide clues for how the presence of bacterial species affects methylation and gene expression, and then induce carcinogenesis.

ANSWER: Thank you for your precious advice.

(1) The data sources of this study from the 338 cancer tissue samples of 375 patients and 69 matched normal tissue samples adjacent to cancer with RNA-seq data, and the cancer tissue samples of 368 patients with HumanMethylation450 data.

(2) A total of 338 patients had RNA sequencing data as well as DNA methylation sequencing data, of which, 19 patients with clinical M stage of M1.

(3) The data types used for each omics in different groups of were summarized in the following table.

Omics	Tumor vs. adjacent control tissues	M0/MX vs. M1	Survival outcome and time
Microbiome analysis	407	338	338
DNA methylation analysis	-	338	338
Bi-mediation analysis	-	338	338

QUESTION 5. The in vitro cell experiment provided by the author was reliable. However, the basis for the selection of verification results was not clearly indicated. Please explain the reasons for verifying the mediation effect between *Staphylococcus saccharolyticus* and *ZNF215* gene.

ANSWER: Thank you for raising this insight problem.

Based on the feasibility of experiment operator and clinical significance, we selected to verify the bioinformatics analysis results of *Staphylococcus saccharolyticus* and *ZNF215* gene. The specific reasons are as follows:

(1) Many studies used several cell experiments. such as CCK-8 cell proliferation experiment, cell clone formation experiment and transwell migration experiment, to reflect

the distant invasion and cloning ability of cells, which was the basis of distant metastasis of cancer cells(10, 11). Therefore, we choose the results of bi-directional mediation analysis results related to distant metastasis of cancer cells and used CCK-8 cell proliferation experiment, cell clone formation experiment for verification.

(2) The bi-directional mediation analysis revealed that three of the twenty potential mediation linkages related to distant metastasis of STAD were related to abnormal methylated genes and bacteria. At present, most studies tended to regard bacteria as the cause of abnormal genetic changes(3-5). The results of data analysis showed that the bi-directional mediation effect between *ZNF215* gene and *S. saccharolyticus* would affect the distant metastasis of cancer cells, so we further determine the direction of effect between them by using cell experiments.

(3) The selection of *ZNF215* gene and *S. saccharolyticus* could provide clues for clinical treatment. *Staphylococcus* was also the dominant bacteria of stomach microbiota, and it was urease-producing organisms. Now, it has been identified that some species of *Staphylococcus* were prevalent in patients with dyspepsia. LEfSe results showed that the relative abundance of *S. saccharolyticus* increased in patients with metastasis. By deleting some virulence genes of *S. saccharolyticus* could be targeted to the related genes in tumor cells to regulate the growth of cancer cells. And previous studies have shown that the *ZNF215* genes can be used as a diagnostic and prognostic biomarker for the Basal-Like Breast Cancer and Acute myeloid leukemia(6, 7). Differential analysis of host DNA methylation showed that specific regions of *ZNF215* gene in metastatic group were hypermethylated and negatively correlated with gene expression level. Epigenetic therapies can also be targeted to specific genomic loci to effectively activate *ZNF215* gene.

-----End of Reply to Reviewer #2-----

Once again, we appreciate the reviewers' insightful suggestions and agree that these will be highly beneficial to the quality of our manuscript. We would like to thank the referee again for taking the time to review our manuscript.

Sincerely
Lei Zhang

1. Sepich-Poore GD, Zitvogel L, Straussman R, Hasty J, Wargo JA, Knight R. 2021. The microbiome and human cancer. *Science (New York, NY)* 371.
2. Li K, Du H, Lian X, Chai D, Li X, Yang R, Wang C. 2016. Establishment and characterization of a metastasis model of human gastric cancer in nude mice. *BMC Cancer* 16:54.

3. Xia X, Wu WKK, Wong SH, Liu D, Kwong TNY, Nakatsu G, Yan PS, Chuang Y-M, Chan MW-Y, Coker OO, Chen Z, Yeoh YK, Zhao L, Wang X, Cheng WY, Chan MTV, Chan PKS, Sung JJY, Wang MH, Yu J. 2020. Bacteria pathogens drive host colonic epithelial cell promoter hypermethylation of tumor suppressor genes in colorectal cancer. *Microbiome* 8:108.
4. Thomas SP, Denu JM. 2021. Short-chain fatty acids activate acetyltransferase p300. *eLife* 10:e72171.
5. Tomlinson MS, Lu K, Stewart JR, Marsit CJ, O'Shea TM, Fry RC. 2019. Microorganisms in the Placenta: Links to Early-Life Inflammation and Neurodevelopment in Children. *PLoS ONE* 14:e020103-18.
6. Maind A, Raut S. 2019. Identifying condition specific key genes from basal-like breast cancer gene expression data. *Computational Biology and Chemistry* 78:367-374.
7. Yang M-Y, Lin P-M, Yang C-H, Hu M-L, Chen IY, Lin S-F, Hsu C-M. 2021. Loss of ZNF215 imprinting is associated with poor five-year survival in patients with cytogenetically abnormal-acute myeloid leukemia. *Blood Cells, Molecules & Diseases* 90:102577.
8. Poore GD, Kopylova E, Zhu Q, Carpenter C, Fraraccio S, Wandro S, Kosciolk T, Janssen S, Metcalf J, Song SJ, Kanbar J, Miller-Montgomery S, Heaton R, McKay R, Patel SP, Swafford AD, Knight R. 2020. Microbiome analyses of blood and tissues suggest cancer diagnostic approach. *Nature* 579:567-574.
9. Tian Y, Morris TJ, Webster AP, Yang Z, Beck S, Feber A, Teschendorff AE. 2017. ChAMP: updated methylation analysis pipeline for Illumina BeadChips. *Bioinformatics*

(Oxford, England) 33:3982-3984.

10. Wang J, Zhang Y, Song H, Yin H, Jiang T, Xu Y, Liu L, Wang H, Gao H, Wang R, Song J. 2021. The circular RNA circSPARC enhances the migration and proliferation of colorectal cancer by regulating the JAK/STAT pathway. *Molecular Cancer* 20:81.
11. Ma C, Wang X, Yang F, Zang Y, Liu J, Wang X, Xu X, Li W, Jia J, Liu Z. 2020. Circular RNA hsa_circ_0004872 inhibits gastric cancer progression via the miR-224/Smad4/ADAR1 successive regulatory circuit. *Molecular Cancer* 19:157.

May 6, 2023

Prof. Lei Zhang
Shandong University
Department of Biostatistics
Jinan
China

Re: Spectrum00904-23R1 (Bi-directional mediation effects between intratumoral microbiome and host DNA methylation changes contribute to Stomach Adenocarcinoma)

Dear Prof. Lei Zhang:

Your manuscript has been accepted, and I am forwarding it to the ASM Journals Department for publication. You will be notified when your proofs are ready to be viewed.

Sincerely,

Zhenjiang Xu
Editor, Microbiology Spectrum
